# Analysis of the Online Reputation Based on Customer Ratings of Lodgings in Tourism Destinations

**Manuel Rodríguez-Díaz \* ![ID], Rosa Rodríguez-Díaz and Tomás F. Espino-Rodríguez ![ID]**

Department of Economics and Business, University of Las Palmas de Gran Canaria, 35017 LasPalmas, Spain;
rosa.rodriguezdiaz@ulpgc.es (R.R.-D.); tomasfrancicos.espino@ulpgc.es (T.F.E.-R.)

\* Correspondence: manuel.rodriguezdiaz@ulpgc.es; Tel.: +34-928-452805

**Abstract:** This study analyzes customers' online social communication to rate lodgings and tourist destinations. A practical methodology is proposed to analyze the online reputation of lodgings as well as the main concepts rated by customers in their online social communication process. To this end, an empirical study was carried out by analyzing the online opinions expressed by customers on the Booking.com tourist lodging website. Based on the information available, three new variables were created and analyzed that represent clearly defined concepts in the minds of consumers. This includes "service quality," "perceived value," and "added value." This study shows that perceived value and service quality are concepts evaluated by customers that are able to differentiate between lodgings in tourist destinations. Therefore, the results show that the online social communication that takes place through this portal has the capacity to guide potential customers by differentiating between the services offered by lodging companies.

**Keywords:** social media; online social communication; tourism destination; online reputation; lodging

## 1. Introduction

The development of digital technology in tourism is influencing the communication and image of tourism companies and destinations (Law et al. 2014; Rodríguez-Díaz et al. 2015; Kim and Park 2017). Customers' online opinions and evaluations of tourist accommodations on specialized websites are determining a new form of intercommunication that creates companies' online reputation (Ye et al. 2014). Social media accounts make it easier for people to share information about their experiences, evaluations, and emotions in a way that influences the decisions of potential clients (Chan and Guillet 2011).

Social media in tourism plays a decisive role on different platforms such as opinion websites (e.g., Booking.com, TripAdvisor, HolidayCheck, Expedia.com, and Hotels.com), social networks (e.g., Facebook, Instagram, Twitter, and YouTube), and opinion leaders' blogs by influencing potential customers' decisions (Kim and Park 2017). These channels are used to share experiences, emotions, conversations, evaluations, images, and information in general (Chan and Guillet 2011) and they represent a new means to promote word-of-mouth communication through the Internet (E-WOM) (Filieri and McLeay 2014; Chen et al. 2011; Leung et al. 2013; Chevalier and Mayzlin 2006; Litvin et al. 2008).

The experiences shared by clients in the network generate a flow of opinions that determine the online reputation and image of lodgings and tourist destinations and affect clients' decision-making behaviors (Liu et al. 2013; Park and Allen 2013; Rodríguez-Díaz et al. 2018a). This new term known as the online reputation is a set of opinions, experiences, and information shared by users of a given product, brand, or service through the social media available on the Internet

(Hernández Estárico et al. 2012). Therefore, the online reputation can be considered a communication activity carried out by customers in an external environment outside the direct control of companies. These communication means make it possible for an inter-relationship to exist between customers and companies in order to reduce the effects of negative opinions and motivate favorable ratings of their products, services, brands, and image (Gössling et al. 2016; Rodríguez-Díaz et al. 2018b).

Online social communication is based on customers' shared opinions and assessments that can be quantitative or qualitative. This study will focus on analyzing the quantitative information available on the Booking.com website about tourist lodgings in three tourist destinations. These assessments focus on measuring the basic constructs for evaluating this type of business such as perceived value (Parasuraman et al. 1988) and the quality of service received by customers (Rust and Oliver 1994; Sweeney and Soutar 2001). In this context, perceived value is a concept that implies the evaluation of service quality, which is directly related, and price, which is inversely related (Rodríguez-Díaz et al. 2015; Rodríguez-Díaz and Espino-Rodríguez 2018a, 2018b).

Value creation is a key concept in management and marketing theories because people identify and evaluate it when making their purchasing decisions. In fact, it is considered a source of competitive advantage for companies to the extent that they have to be oriented towards satisfying the desires and needs of customers (Porter 1980; Barney 1991; Grant 1991; Bharadwaj et al. 1993; Grönroos 2007; Payne and Frow 2005; Ngo and O'Cass 2009). The concepts of value, value creation, and perceived value are considered by the American Marketing Association (AMA) in their proposed definition of marketing. The AMA establishes that these notions refer to the activity, the framework of institutions, and the processes for creating, communicating, making, and exchanging offers that have value for customers, partners, and societies (AMA 2007).

In this new digital world, it is necessary to analyze clients' social media as well as the evaluations made of a specific brand, company, or destination such as in the case of the tourism industry. Based on this, the objectives to be achieved in this study are the following: (1) Analyze the customers' quantitative evaluations of tourist lodgings in relation to the basic concepts of the quality of service, the perceived value, and added value based on the information available on the website of Booking.com and determine whether the social communication process makes it possible to detect significant differences between tourist destinations, which will be analyzed.

To achieve these goals, we begin by reviewing the literature concerning an online reputation and the main constructs that are evaluated quantitatively. The methodology applied is set out below. With this, the results obtained are presented in the following section. The study ends with a discussion of the main results obtained as well as the conclusions.

## 2. Literature Review

The emergence of social media in everyday life is bringing about a profound change in people's communication processes (Kim and Park 2017). The exchange of opinions, comments, experiences, points of view, photos, and videos allows people to create an opinion about a product, service, company, or brand. Different types of platforms are specialized in promoting this new method of communication, which can be individual or mass. They include social networks and specialist opinion websites and blogs where opinion leaders influence the creation of opinion flows as well as people's buying behavior (Chan and Guillet 2011). In this context, a new term called E-WOM refers to the process of word-of-mouth communication where one consumer influences another potential consumer through shared online ratings and opinions (Filieri and McLeay 2014; Chen et al. 2011; Leung et al. 2013; Chevalier and Mayzlin 2006; Litvin et al. 2008).

Mauri and Minazzi (2013) list the main differences between the traditional concept of word-of-mouth (WOM) communication and the E-WOM. First, electronic communication is not necessarily direct or oral since users often write their opinions or carry out surveys at pre-determined scales on websites. Second, the communication remains on the network and can be consulted for a long period of time but cannot be considered as advertising. Third, the information shared may

be about a brand, company, product, service, or destination (Hernández Estárico et al. 2012). Lastly, WOM communication generates a greater degree of trust than E-WOM because the transmission of personal WOM feedback is spontaneous and based on experience. On the contrary, communication via the Internet can raise doubts about its veracity since companies themselves can intervene to influence an online reputation (Yacouel and Fleischer 2012). There may also be biases in customer opinions or tendentious interference by competitors (Rodríguez-Díaz et al. 2018b).

Ladhari and Michaud (2015) point out that the E-WOM assumes that users want to share their assessments socially. To the extent that potential customers give a level of truthfulness to these comments, this will directly influence hotel sales (Wahab et al. 2015). In this context, Inversini et al. (2009) and Micera and Crispino (2017) point to the influence that the information available on the Internet has on the level of competitiveness of tourist destinations. Moreover, Micera and Crispino (2017) propose a methodological framework for analyzing the process of developing the image of tourist destinations and integrating sentiment analysis within the social network analysis tools and social media analytics. Due to Internet communication, the study of the impact of web sites of accommodation is becoming increasingly important (Law and Hsu 2005). Specialized websites such as Booking.com, HolidayCheck, or TripAdvisor have achieved great relevance and have replaced, to a considerable degree, the specific websites of each accommodation. It is important to point out that the strategy of direct customer capturing through the web page is a competitive advantage for the accommodation firms as they can obtain more information from the clients (Jeong et al. 2003), develop loyalty strategies, have the possibility of obtaining a higher margin of income by eliminating intermediaries in the channel, and carry out promotional actions (Rong et al. 2009).

A company's online reputation is shaped by the quantitative and qualitative information available in specialized databases that directly influence the outcome of accommodations (Luca 2011; Noone et al. 2011; Ye et al. 2009; Varini and Sirsi 2012; Anderson 2012). Torres (2014) states that comments are qualitative evaluations that measure the level of customer satisfaction while the quality of service perceived is usually measured by quantitative variables. Quantitative evaluations are based on scales of variables that measure specific concepts, which, due to their great impact on purchasing behavior, have been widely studied in the academic literature (Ye et al. 2014; Rodríguez-Díaz and Espino-Rodríguez 2018a, 2018b). Scales aimed at measuring opinions about tourist lodgings generally focus on two basic concepts: perceived value and the quality of service. However, Rodríguez-Díaz et al. (2015) developed a method to measure value added from these two constructs. Therefore, the social communication carried out on the Internet about key aspects of a service, product, brand, or company is based on constructs defined and analyzed in specialized academic literature, which usually measures the attitudes and perceptions of customers (Parasuraman et al. 1988; Grönroos 2007).

The concept of value is identified by customers in order to evaluate the products or services they wish to contract and it is an essential construct for the development of companies' strategies (Porter 1980; Barney 1991; Grant 1991; Rumelt 1984, 1991; Wernerfelt 1984; Grönroos 2007). The greater the value perceived by clients, the greater the results and profitability, which are also subject to appropriate cost management (Gale 1994). The objective of companies should be to develop a long-term competitive value proposition that is perceived as such by their customers (Day 1990; Slater and Narver 1994; Slater 1997; Woodruff 1997).

To define business strategies and tactics within the service sector, value creation is an essential aspect given the subjectivity of the term, which is influenced by customers' circumstances, perceptions, and attitudes (Holbrook 1994; Anderson and Narus 1998). Value has been studied by several authors. Holbrook (1994) defines it as "a relative preference (comparative, personal, situational) characterized as an experience of a subject interacting with an object." Zeithaml (1988) carries out a study with the aim of specifying the possible definitions of the concept of value and arrive at the conclusion that there are four variables: (a) value is low price, (b) value is what the client wants from a product, (c) value is

the quality that a client obtains for the price paid, and (d) value is what the client obtains for what s/he gives in return.

Naumann (1995) states that there are three dimensions that define value: product quality, service quality, and value based on price. This author considers that the customer value is created at the exact moment when the company's offer is equal to or greater than the expectations created by the three previous factors. On the other hand, Rust and Oliver (1994, p. 10) state that "value is made up of perceived quality in combination with price." They also view value as the difference between the usefulness of quality and the uselessness of price.

Different methods have been applied in the academic literature to measure perceived value and quality. There has also been considerable discussion about how to identify the relationships among the constructs of service quality, service value, and customer satisfaction (Cronin et al. 2000; Sweeney and Soutar 2001; Ulaga and Eggert 2006). In scientific marketing research, different studies have been conducted on customer satisfaction (Oliver 1997) and expected and perceived quality (Parasuraman et al. 1988) as well as how to measure customers' perceived value (Rust and Oliver 1994; Sweeney and Soutar 2001; Rodríguez-Díaz and Espino-Rodríguez 2018a). From a more operational perspective, most studies have focused on determining client satisfaction (Chadee and Mattsson 1996; Baker and Crompton 2000; Füller et al. 2006; Nam et al. 2011) and service co-creation (Prahalad and Ramaswamy 2004; Cabiddu et al. 2013). By contrast, few studies have focused on the creation of value, added value, and competitive positioning in the tourist accommodation market (Williams and Soutar 2009; Tajzadeh-Namin 2012; Rodríguez-Díaz et al. 2015, 2018a).

In recent years, online customer reviews have been used in scientific tourism research (Ye et al. 2014). Authors such as Mudambi and Schuff (2010) define online surveys as evaluations of the products and services received by customers, which, in turn, are transmitted directly to companies or through a third party such as websites. In this area, online customer evaluations have become one of the main sources of information for evaluating the quality of products and services (Hu et al. 2008).

In the services industry, online assessments are considered a very useful tool for investigating customer perceptions (Pantelidis 2010; Ryu and Han 2010; Vermeulen and Seegers 2009; Zhang et al. 2010) and developing a strategic analysis to improve the competitiveness of tourism businesses. So Ye et al. (2014, p. 36) state that hotel managers should seriously consider using online customer ratings when defining pricing strategies.

In the field of tourism, the scientific research has already accepted that, compared to conventional questionnaires and interviews, online evaluations provide a new and effective way to investigate customer perceptions not only for researchers but also for managers and directors seeking practical applications applied to tourism (Ye et al. 2014; Prebensen et al. 2012; Rodríguez-Díaz et al. 2015; Rodríguez-Díaz and Espino-Rodríguez 2018a, 2018b). In addition, this methodology, despite being based on scales made up of a small number of variables, is satisfactory for determining the level of clients' evaluations of tourist lodgings and destinations.

## 3. Methodology

This empirical study is based on the creation of a database using the information available on Booking.com. Client evaluations have been selected for 272 resorts in the south of Gran Canaria (Spain, Canary Islands), 83 in the south of Tenerife (Spain, Canary Islands), and 49 in Agadir (Morocco). Therefore, a total of 404 tourist lodgings have been considered for statistical analysis. The number of customers who shared their opinions and assessments through the website studied totals 69,030 of which 38,102 opinions corresponded to accommodations in Gran Canaria, 20,950 to Tenerife, and the rest (9978 opinions) to complexes located in Agadir.

Both Gran Canaria and Tenerife are among the most attractive tourist destinations on the Canary Islands while Agadir is considered a direct competitor due to its geographical proximity and similar climate. The Canary Islands receive 12 million tourists a year (ISTAC 2015) of which four million tourists visit Gran Canaria and five million tourists visit the island of Tenerife. On the other

hand, Agadir and the Souss Massa Drâa region receive approximately four million tourists per year (ICEX 2011). Both destinations are specialized in sun and beach tourism because they are located very close to each other in the middle of the Atlantic Ocean. The distance between the Canary Islands and Agadir is 420 km.

The online tourism portal Booking.com has been chosen to carry out this study because it offers opinions based on the experiences of real clients and, therefore, it has been considered a powerful source of reliable and adequate information for study. In addition, other tourism opinion platforms do not use such exhaustive control measures to verify that the evaluations belong to real clients, which reduces their levels of statistical validity (Rodríguez-Díaz and Espino-Rodríguez 2018b).

On Booking.com, a survey is available to users based on six scoring questions with a 10-point scale (1 = very poor rating and/or experience, 10 = very good rating and/or experience). However, Mellinas et al. (2015) clarified that the scale used by Booking.com actually has four points, which means it is then transformed into a 10-point score. In this context, these authors demonstrate that, in reality, the scale ranges from a minimum score of 2.5 to a maximum score of 10 points.

The scores apply to the variables of cleanliness (Li), comfort (C), location (Lo), facilities and services (I), personnel (P), value for money (V), and Wi-Fi (W). Booking.com calculates the average of these variables to obtain an overall value of the accommodation, which it calls the "hotel average." Other data available in this portal are the category and prices of lodgings, which will also be analyzed in this study. Therefore, using the available information (see Table 1), Booking.com calculates an average hotel score (PH) using the following formula (Rodríguez-Díaz et al. 2015).

$$AVGH = (S + F + Cl + Co + L + V + W)/6$$

**Table 1.** Description of the variables.

| Variables | Description |
|---|---|
| Average of the hotel (AVGH) | Average of all variables evaluated by customers of the lodging |
| Staff (S) | Global evaluation of the staff of the lodging |
| Facilities (F) | Global evaluation of the facilities and service of the lodging |
| Cleanliness (Cl) | Evaluation of the cleanliness of the lodging |
| Comfort (Co) | Evaluation of the global comfort of the lodging |
| Location (L) | Global evaluation of the location of the lodging |
| Value for money (V) | Evaluation of the perceived value by customers |
| Price | Price per night and double room |
| Category | Number of stars or keys |

On the one hand, to determine the value for the client, there is a specific question on Booking.com that measures the "value for money." Different authors treat it differently from the rest of the variables in order to study the clients' evaluations of the perceived value of the tourist accommodations (Ye et al. 2014; Rodríguez-Díaz et al. 2015). On the other hand, a new variable called 'quality of service' (Q) is calculated through the mean of the other variables to measure this construct. We should clarify that the Wi-Fi variable is not included because it depends to a large extent on the public infrastructures of each destination and, in addition, it is a variable included relatively recently. Therefore, the quality of service Q is determined by using the following formula.

$$Q = (S + F + Cl + Co + L)/5$$

"Added value" (AV) is another variable that can be analyzed using the method proposed by Rodríguez-Díaz et al. (2015). This variable represents what a tourist accommodation offers, more or

less, in terms of value for money. In this context, AV is calculated from the difference between 'value for money' or the 'quality-price relationship' and 'quality' (Q).

$$AV = V - Q$$

When a customer chooses a lodging with high quality and price, it is possible that s/he gets a negative AV because his/her expectation is higher than the price paid. Conversely, a customer who buys a cheap but medium quality accommodation may perceive a positive AV. This new variable introduced in the study can take three types of values, which determines a competitive positioning of tourism businesses as follows: (1) VA = 0, when customers consider the quality offered by the tourist lodging to be consistent with its price, (2) VA > 0, when customers consider the price they have to pay to be lower than the quality of the service received from the tourist lodging and (3) VA < 0 when customers consider the price to be higher than the quality of the service received from the tourist lodging.

## 4. Analysis of Results

In order to obtain the results of the research, different statistical analyses have been carried out. The first consists of a descriptive analysis of the indicators that determine the general characteristics of the accommodation studied. Second, a cross-tables analysis is carried out to compare the results obtained for the lodgings in the tourist destinations. Third, an ANOVA analysis is applied in order to look for significant differences between the averages of each of the variables, according to the destinations. However, before starting with our analyses, we will describe the particular characteristics of each destination in terms of the types of lodging offered and their categories.

### 4.1. Characteristics of Lodgings by Destinations

The general characteristics of the accommodations analyzed include their type and category. To this end, a series of histograms has been drawn up reflecting the results obtained in terms of the type of lodging and category for Gran Canaria, Tenerife, and Agadir. In the histograms referring to the type of accommodation (see Figures 1–3), the value of 0 corresponds to apartments. The value of 1 to bungalows, the value of 2 to aparthotels, and the value of 3 to hotels. As for the category, the value coincides with the number of stars or keys of the accommodation.

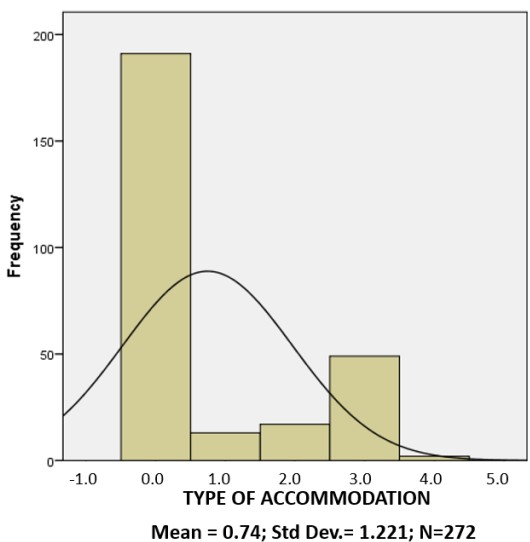 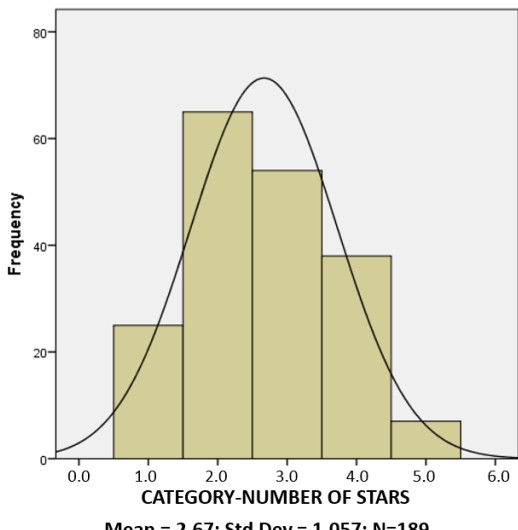

**Figure 1.** Types and categories of accommodation in the south of Gran Canaria.

Figure 1 shows that, in the accommodations studied in Gran Canaria, there is a greater number of apartments, which is followed by hotels. In addition, most of them are classified as category two and three and are followed by the category of four stars. Regarding category five accommodations, there are very few, according to the data available on Booking.com because not all the resorts have been catalogued.

In the case of Tenerife, the figure shows that apartments are a little more prevalent, but with very little difference. In relation to the category, three and four stars or keys have the same frequency, which means that the accommodations offered in Tenerife have higher categories than those offered on the island of Gran Canaria. The curve presented by the category variable in Figure 2 is nearly uniform because categories two and five also present very similar frequencies.

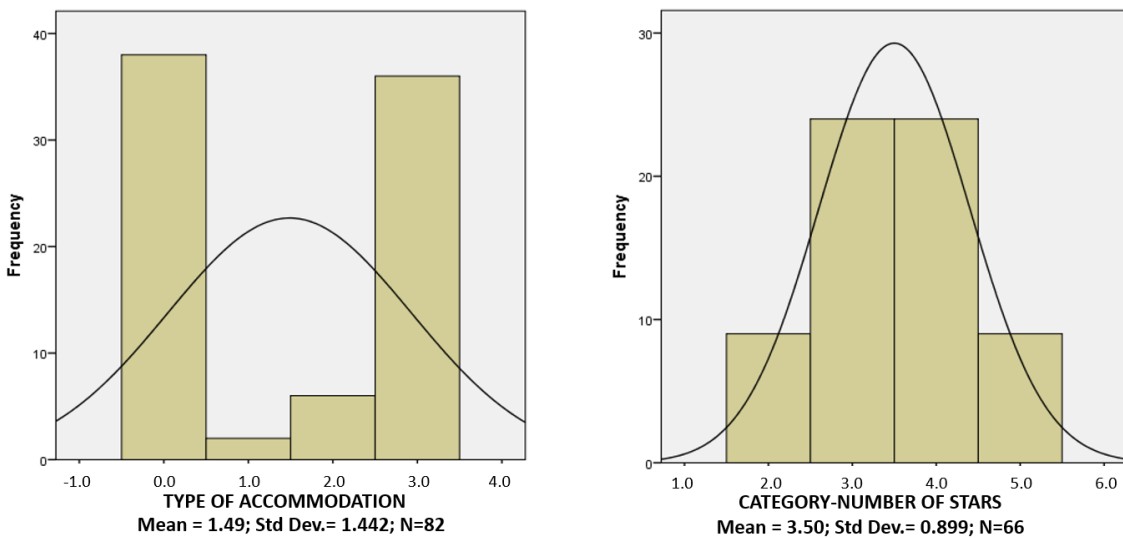

**Figure 2.** Types and categories of accommodation in the south of Tenerife.

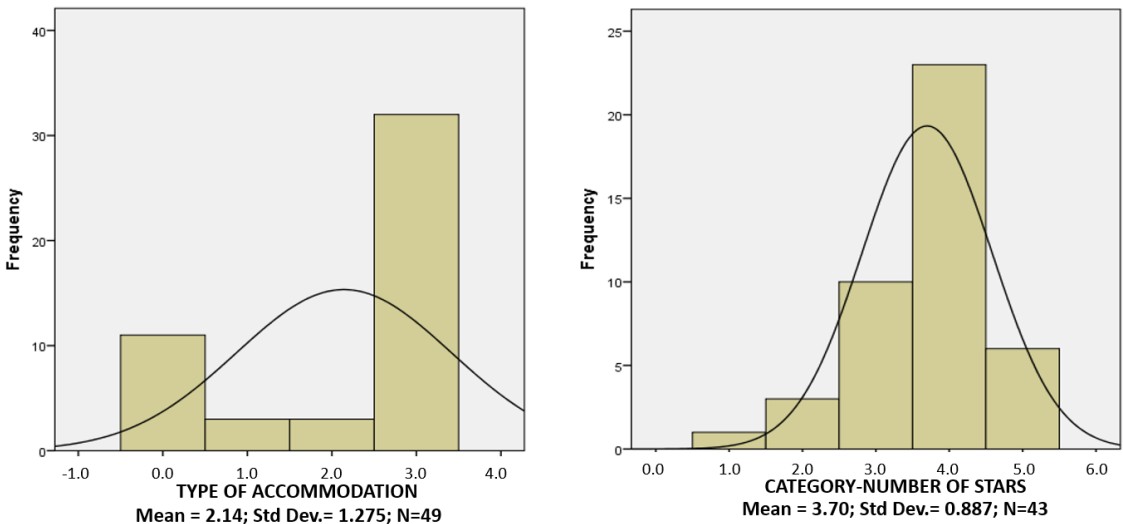

**Figure 3.** Types and categories of accommodation in Agadir.

On the other hand, Agadir presents quite different results from Gran Canaria (see Figure 3) due to the fact that the accommodations offered by Booking.com in this destination are mostly hotels with a four-star category or keys. Thus, by analyzing these variables, the quality of Agadir and its added value should be higher, but the statistical analyses carried out later will allow us to verify whether this is the case based on the customers' perceptions.

*4.2. Comparative Analysis of Destinations*

The cross-tables analysis is developed in this section in order to describe the results obtained in the three destinations studied. In order to clarify the presentation of the results, the variables have been re-coded in a smaller range from a scale of 10 to one with 5 alternatives, which differentiates the variable 'added value' from the rest because it usually moves over the value 0 with positive or negative results. Therefore, the variables that measure service quality and value were re-coded by intervals (see below).

- Until 5.9 value 0 → Very low evaluation.
- From 6 to 6.9 value 1 → Low evaluation.
- From 7 to 7.9 value 3 → Average evaluation.
- From 8 to 8.9 valor 4 → High evaluation.
- From 9 to 10 valor 5 → Very high evaluation.

The added value variable, because it can have positive and negative values, was recoded in the following way.

- From −10 to −1 value 0 → Very low evaluation.
- From −0.99 to −0.5 value 1 → Low evaluation.
- From −0.49 to 0 value 3 → Average evaluation.
- From 0.01 to 0.5 value 4 → High evaluation.
- From 0.51 to 10 value 5 → Very high evaluation.

Table 2 shows that there are significant differences between destinations in terms of cleanliness. Thus, 8.2% of the customers who visit Agadir give the cleaning a very low evaluation compared to Gran Canaria and Tenerife, which only reach 2.6% and 1.2%, respectively, for the lowest scale evaluation. On the other hand, Tenerife is the destination with the highest percentage of customers who give a very high rating (18.3%) followed by Gran Canaria (15.0%) while only 6.1% of Agadir's accommodations get the best ratings from their customers.

**Table 2.** Comparative analysis of the cleanliness variable.

| Cleanliness | Destinations | | | |
| --- | --- | --- | --- | --- |
| Intervals | Gran Canaria | Tenerife | Agadir | Total |
| Very low evaluation | 2.6% | 1.2% | 8.2% | 3.0% |
| Low evaluation | 11% | 17.1% | 30.6% | 14.6% |
| Average evaluation | 36.3% | 34.1% | 34.7% | 35.6% |
| High evaluation | 35.2% | 29.3% | 20.4% | 32.2% |
| Very high evaluation | 15% | 18.3% | 6.1% | 14.6% |
| Total | 100% | 100% | 100% | 100% |

A similar trend can be seen for comfort with 16.3% of Agadir's customers rating it as very low and 34.7% as low (see Table 3). Despite the fact that the results obtained in Gran Canaria and Tenerife are very similar, the percentages obtained by this latter destination stand out with 18.3% reaching the highest score, but only 7.0% in Gran Canaria. This may be because Tenerife's accommodation offer is more hotel-oriented and more modern than Gran Canaria's.

**Table 3.** Comparative analysis of the comfort variable.

| Comfort | Destinations | | | |
| --- | --- | --- | --- | --- |
| Intervals | Gran Canaria | Tenerife | Agadir | Total |
| Very low evaluation | 5.9% | 2.4% | 16.3% | 16.3% |
| Low evaluation | 25.3% | 23.2% | 34.7% | 26.0% |
| Average evaluation | 41.0% | 39.0% | 28.6% | 39.1% |
| High evaluation | 20.9% | 17.1% | 18.4% | 19.8% |
| Very high evaluation | 7.0% | 18.3% | 2.0% | 8.7% |
| Total | 100% | 100% | 100% | 100% |

Location is another essential variable in determining the level of service offered by accommodations to their customers. In both Gran Canaria and Tenerife, customers rated the location as high. The opposite is true in Agadir where 2% rate the location of the lodgings as very low. However, Table 4 shows that, in general, the location is considered positive regardless of the destination analyzed. Gran Canaria stands out because 24.5% rate the location as very high.

**Table 4.** Comparative analysis of the location variable.

| Location | Destinations | | | |
| --- | --- | --- | --- | --- |
| Intervals | Gran Canaria | Tenerife | Agadir | Total |
| Very low evaluation | 0.0% | 0.0% | 2.0% | 0.2% |
| Low evaluation | 5.1% | 7.3% | 8.2% | 5.9% |
| Average evaluation | 23.8% | 23.2% | 34.7% | 25.0% |
| High evaluation | 46.5% | 51.2% | 53.1% | 48.3% |
| Very high evaluation | 24.5% | 18.3% | 2.0% | 20.5% |
| Total | 100% | 100% | 100% | 100% |

The results achieved for the facilities and services variable are shown in Table 5. The table shows that the destination of Agadir obtained very poor ratings since 20.4% of the customers consider it to be very low, 38.8% low, and 30.6% of medium quality. Only 10.2% of the customers give it the highest ratings on the scale. Tenerife also obtained slightly better results than Gran Canaria especially on the maximum evaluation where it obtained 11% when compared to 4.8% in Gran Canaria.

**Table 5.** Comparative analysis of the facilities variable.

| Facilities | Destinations | | | |
| --- | --- | --- | --- | --- |
| Intervals | Gran Canaria | Tenerife | Agadir | Total |
| Very low evaluation | 7.0% | 2.4% | 20.4% | 7.7% |
| Low evaluation | 26.0% | 23.2% | 38.8% | 27.0% |
| Average evaluation | 41.4% | 41.5% | 30.6% | 40.1% |
| High evaluation | 20.9% | 22.0% | 8.2% | 1.6% |
| Very high evaluation | 4.8% | 11.0% | 2.0% | 5.7% |
| Total | 100% | 100% | 100.0% | 100.0% |

Regarding the personnel variable, Table 6 shows that the results are, in general, more homogeneous with the exception of Agadir where 32.7% of customers consider that the service offered by the staff is low quality. In this case, we can also appreciate greater attention to this variable in the companies located in Gran Canaria and Tenerife because it is an essential aspect of the quality of service perceived by customers.

**Table 6.** Comparative analysis of the staff variable.

| Staff | Destinations | | | |
|---|---|---|---|---|
| Intervals | Gran Canaria | Tenerife | Agadir | Total |
| Very low evaluation | 0.4% | 1.2% | 4.1% | 1.0% |
| Low evaluation | 5.5% | 2.4% | 32.7% | 8.2% |
| Average evaluation | 29.7% | 42.7% | 34.7% | 32.9% |
| High evaluation | 47.6% | 37.8% | 20.4% | 42.3% |
| Very high evaluation | 16.8% | 15.9% | 8.2% | 15.6% |
| Total | 100% | 100% | 100% | 100% |

With the variables previously analyzed, a new variable Q has been created that is determined by averaging the evaluations obtained for the five variables described. Table 7 shows that no lodgings in Agadir have the highest rating for quality of service. By contrast, in Tenerife, 12.2% obtained this rating and 7.7% obtained the rating in Gran Canaria. Among the lowest ratings, the fact that 34.7% of Agadir's customers consider the quality of the service offered by the accommodation to be low is noteworthy.

**Table 7.** Comparative analysis of the quality (Q) variable.

| Quality (Q) | Destinations | | | |
|---|---|---|---|---|
| Intervals | Gran Canaria | Tenerife | Agadir | Total |
| Very low evaluation | 1.5% | 1.2% | 4.1% | 1.7% |
| Low evaluation | 12.1% | 8.5% | 34.7% | 14.1% |
| Average evaluation | 43.4% | 48.8% | 40.8% | 44.2% |
| High evaluation | 35.3% | 29.3% | 20.4% | 32.3% |
| Very high evaluation | 7.7% | 12.2% | 0.0% | 7.7% |
| Total | 100% | 100% | 100% | 100% |

The perceived value variable is another key concept that helps understand customer behavior and provides useful assessments of tourism services. It is a variable that is directly conditioned by the quality of the service and inversely by the price paid. Table 8 shows that almost 50% of Agadir's lodgings are assessed by its clients as having low or very low value. On the other hand, in Gran Canaria and Tenerife, the ratings are mostly concentrated in the medium and high ratings, which shows that customers consider the service they receive to be adequate for the price they pay.

**Table 8.** Comparative analysis of the value for the money variable.

| Value for Money | Destinations | | | |
|---|---|---|---|---|
| Intervals | Gran Canaria | Tenerife | Agadir | Total |
| Very low evaluation | 1.1% | 0.0% | 6.1% | 1.5% |
| Low evaluation | 13.2% | 17.1% | 44.9% | 17.8% |
| Average evaluation | 48.7% | 46.3% | 28.6% | 45.8% |
| High evaluation | 35.9% | 34.1% | 18.4% | 33.4% |
| Very high evaluation | 1.1% | 2.4% | 2.0% | 1.5% |
| Total | 100% | 100% | 100% | 100% |

The added value is a variable determined by subtracting the quality of the service offered by the accommodation from the perceived value (Rodríguez-Díaz et al. 2015). In this case, the results are more homogeneous with 12.25% of the lodgings in Agadir obtaining the highest score, i.e., the quality of the service received is considered by the clients to be higher than the price paid (see Table 9). Curiously, the worst results are also obtained by Agadir because 12.25% of its accommodations are evaluated

with a very low added value. This means that the price paid by customers is significantly higher than the quality of the service obtained.

**Table 9.** Comparative analysis of the added value variable.

| Added Value | Destinations | | | |
|---|---|---|---|---|
| Intervals | Gran Canaria | Tenerife | Agadir | Total |
| Very low evaluation | 6.96% | 7.32% | 12.25% | 7.67% |
| Low evaluation | 13.55% | 31.71% | 20.41% | 18.07% |
| Average evaluation | 33.70% | 20.73% | 20.41% | 29.46% |
| High evaluation | 33.90% | 30.49% | 34.69% | 34.65% |
| Very high evaluation | 9.89% | 9.76% | 12.25% | 10.15% |
| Total | 100% | 100% | 100% | 100 % |

The analysis carried out indicates that the communication by the clients of tourist accommodations through the Internet discriminates between the destinations analyzed. This means that customers give opinions and evaluations that really make a difference in order to help potential customers. Based on these results, it can be said that online social communication about tourist lodgings is a phenomenon that really generates value for users insofar as it provides information that allows them to compare alternatives. In addition, it identifies destinations that have similar characteristics such as in the case of Gran Canaria and Tenerife and it differentiates between destinations where customers perceive that the services provided are not the same such as in the case of Agadir.

*4.3. ANOVA Analysis*

After presenting the results for each of the variables in the three tourist destinations and determining that there are differences in certain cases, the One-Way ANOVA analysis (Analysis of Variance) was applied with the objective of validating these differences. This statistical method is based on the comparison of averages of quantitative variables referring to a qualitative classification variable, which, in this case, is the tourist destination. First, the results of the overall analysis of the three destinations are presented and then partial analyses are carried out between two destinations in order to establish where the significant differences lie in relation to the variables and the specific destinations.

Table 10 shows the results of applying the ANOVA analysis to all three destinations for each of the variables studied. The critical level of significance, set at 0.05, is only exceeded by the value-added variable (0.312), which, as indicated above, can have positive and negative values and is very subjective because of the effect of price on customers. These results show that online social communication transmitted through the Booking.com website discriminates between destinations on all the variables except one known as the added value, which proves to be an effective process for exchanging information between users.

To examine the results in more depth and determine where the differences between destinations lie, the Scheffe Test was carried out. The results are shown in Table 11. This statistical analysis classifies tourist destinations according to the similarity or difference in the averages obtained. Thus, on the cleanliness variable, the average obtained by Agadir (7.2592) is lower than the one obtained by the destinations of Gran Canaria (7.9128) and Tenerife (7.8939), which make up another group. The same is true for the rest of the variables that measure the quality of service such as comfort, location, facilities and services, and personnel. Again, the destinations of Gran Canaria and Tenerife are similar and maintain a significant competitive difference with respect to Agadir where all the variables are given lower scores by customers. These results confirm that the customer information collected on Booking.com discriminates between tourist destinations, which is essential for guiding future customers.

**Table 10.** ANOVA analysis of the destinations of Gran Canaria, Tenerife y Agadir.

| Variables | Sum of Squares | gl | Squared Average | F | Sig. |
|---|---|---|---|---|---|
| Cleanliness | 18.173 | 2 | 9.087 | 9.768 | 0.000 |
| Comfort | 14.347 | 2 | 7.173 | 7.289 | 0.001 |
| Location | 9.030 | 2 | 4.515 | 7.378 | 0.001 |
| Facilities | 25.961 | 2 | 12.980 | 14.109 | 0.000 |
| Staff | 27.491 | 2 | 13.745 | 20.775 | 0.000 |
| Quality Q | 16.659 | 2 | 8.329 | 13.306 | 0.000 |
| Value for money V | 16.930 | 2 | 8.465 | 17.105 | 0.000 |
| Average of the hotel | 17.277 | 2 | 8.638 | 15.244 | 0.000 |
| Added value AD = V − Q | 0.702 | 2 | 0.351 | 1.167 | 0.312 |
| Minimum price in low season | 31,997.120 | 2 | 15,998.560 | 3.708 | 0.025 |
| Maximum price in low season | 106,250.367 | 2 | 53,125.183 | 15.229 | 0.000 |
| Minimum price in high season | 71,564.044 | 2 | 35,782.022 | 5.704 | 0.004 |
| Maximum price in high season | 85,775.182 | 2 | 42,887.591 | 4.835 | 0.009 |

The generic variable measuring the quality of service Q also shows significant differences in favor of the destinations of Gran Canaria (7.8467) and Tenerife (7.9034) when compared to Agadir (7.2416). The same thing applies to the average score of all the variables on the scale where the lodgings in Agadir obtained an average (7.2061) significantly lower than those of Gran Canaria (7.8318) and Tenerife (7.8609). As indicated above, the value added variable does not show significant differences among the three destinations studied. They are all classified in the same group. As far as the quality/price ratio is concerned, the same result can be seen again where Agadir with an average accommodation score of 7.0571 shows a competitive disadvantage compared to the destinations of Gran Canaria (7.6926) and Tenerife (7.6439).

The analysis of price in high and low season yields very interesting results due to the fact that it discriminates among the three destinations. With regard to the minimum prices in the low season, although the ANOVA analysis gave a significance level of 0.025, i.e., significant at 5%, the Scheffe test grouped the three destinations in the same group. The rest of the prices studied differentiate between destinations especially high prices in the low season where the destinations of Gran Canaria and Agadir are assigned to the same group while Tenerife has significantly higher prices. On the other hand, in the high season, prices in the Canary Islands are significantly higher than those in Agadir, which demonstrates a higher competitive level.

**Table 11.** The Scheffe test of the tourism destinations.

| Cleanliness | | | Hotel Average | | |
|---|---|---|---|---|---|
| | 1 | 2 | | 1 | 2 |
| AGADIR | 7.2592 | | AGADIR | 7.2061 | |
| TENERIFE SUR | | 7.8939 | GRAN CANARIA SUR | | 7.8318 |
| GRAN CANARIA SUR | | 7.9128 | TENERIFE SUR | | 7.8609 |
| Sig. | 1.000 | 0.992 | Sig. | 1.000 | 0.970 |
| COMFORT | | | ADDED VALUE V-Q | | |
| | 1 | 2 | | | 1 |
| AGADIR | 7.0347 | | TENERIFE SUR | | −0.2595 |
| GRAN CANARIA SUR | | 7.4501 | AGADIR | | −0.1845 |
| TENERIFE SUR | | 7.7183 | GRAN CANARIA SUR | | −0.1541 |
| Sig. | 1.000 | 0.222 | Sig. | | 0.467 |

**Table 11.** *Cont.*

| LOCATION | 1 | 2 | MINIMUM PRICE IN LOW SEASON | 1 | |
|---|---|---|---|---|---|
| AGADIR | 7.8796 | | AGADIR | 64.2296 | |
| TENERIFE SUR | | 8.2719 | GRAN CANARIA SUR | 67.7168 | |
| GRAN CANARIA SUR | | 8.3458 | TENERIFE SUR | 89.4855 | |
| Sig. | 1.000 | 0.832 | Sig. | 0.051 | |

| FACILITIES | 1 | 2 | MAXIMUM PRICE IN LOW SEASON | 1 | 2 |
|---|---|---|---|---|---|
| AGADIR | 6.6714 | | GRAN CANARIA SUR | 73.8072 | |
| GRAN CANARIA SUR | | 7.382 | AGADIR | 77.8276 | |
| TENERIFE SUR | | 7.5500 | TENERIFE SUR | | 115.3683 |
| Sig. | 1.000 | 0.531 | Sig. | 0.910 | 1.000 |

| STAFF | 1 | 2 | MINIMUM PRICE IN HIGH SEASON | 1 | 2 |
|---|---|---|---|---|---|
| AGADIR | 7.3633 | | AGADIR | 67.0295 | |
| TENERIFE SUR | | 8.0829 | TENERIFE SUR | | 111.5185 |
| GRAN CANARIA SUR | | 8.1758 | GRAN CANARIA SUR | | 112.6080 |
| Sig. | 1.000 | 0.764 | Sig. | 1.000 | 0.997 |

| QUALITY Q | 1 | 2 | MAXIMUM PRICE IN HIGH SEASON | 1 | 2 |
|---|---|---|---|---|---|
| AGADIR | 7.2416 | | AGADIR | 81.2112 | |
| GRAN CANARIA SUR | | 7.8467 | GRAN CANARIA SUR | | 125.866 |
| TENERIFE SUR | | 7.9034 | TENERIFE SUR | | 137.5617 |
| Sig. | 1.000 | 0.899 | Sig. | 1.000 | 0.779 |

| VALUE FOR MONEY V | 1 | 2 |
|---|---|---|
| AGADIR | 7.0571 | |
| TENERIFE SUR | | 7.6439 |
| GRAN CANARIA SUR | | 7.6926 |
| Sig. | 1.000 | 0.905 |

## 5. Discussion and Conclusions

Online social communication is a process through which users share information, opinions, or experiences that, when focused on a particular product, service, image, or company, develop an online reputation (Hernández Estárico et al. 2012). Through this new phenomenon based on social media, a strong influence is exerted on potential consumers, which develops companies' competitive and communication strategies by using the E-WOM (Ladhari and Michaud 2015; Mauri and Minazzi 2013). When social communication is carried out through tourist opinion websites such as Booking.com, the results are decisive for creating an adequate and competitive online reputation for tourist accommodations and destinations, which is an aspect that is demonstrated in the study (Wahab et al. 2015; Inversini et al. 2009; Micera and Crispino 2017).

This study analyzed the extent to which quantitative information shared by clients is effective and useful based on its ability to differentiate between lodgings in tourist destinations. This is an essential factor because, if the opinions expressed on the network tend to be counterbalanced or very homogeneous, they will not be valuable to users. In this context, the results demonstrate that effective online social communication can detect essential differences that may influence the evaluations made by potential consumers in their decision-making processes (Rodríguez-Díaz and Espino-Rodríguez 2018a). If this characteristic is not present, it can be deduced that the process of

online social communication leads to homogenization of information and, therefore, loses the capacity to influence or provide essential value to the people who receive this information (Jeong et al. 2003).

The comparative study shows that there are differences between destinations. Accommodation in Tenerife has the highest overall rating especially in terms of comfort and facilities. Gran Canaria is the next destination in terms of results with similar results in terms of cleaning and staff, which improves the location. Analyzing the quality variable (Q), we can see that both Gran Canaria and Tenerife obtained similar results in the highest scores while the Agadir evaluations dropped significantly. The value for money is another key variable that relates the quality of service perceived to the price paid by customers (Holbrook 1994) and it is acknowledged that Gran Canaria and Tenerife obtain significantly better results than Agadir. On the other hand, in terms of added value (Rodríguez-Díaz et al. 2015), Agadir is the one that achieves the best rates since it has lower prices than the two Canary Islands destinations.

The only variable that shows no significant difference across the three destinations is the added value ($p > 0.05$). In addition, the minimum prices in the low season, despite obtaining a level of significance of 0.025 in the ANOVA analysis, were assigned by the Scheffe test to the same group. Among all the service quality variables as well as the aggregation variables such as the Q quality and the average score ($p < 0.05$), the destinations of Gran Canaria and Tenerife were assigned to the same group, which shows significant differences from the accommodations in the Agadir destination. The only exception was the minimum price during the low season where Gran Canaria and Agadir were classified in the same group due to the fact that their prices are significantly lower than those of Tenerife.

Therefore, the statistical analyses carried out have shown that the information on customer ratings collected on Booking.com differentiates between destinations (Rodríguez-Díaz and Espino-Rodríguez 2018a, 2018b). This is a great relief for social communication because it is a way of contrasting the degree of validity of the information shared on online portals. If social communication is homogeneous when evaluating substitute products, services, or brands, it does not generate value for people who use it to make decisions. In this context, it is important that customers' online evaluations offer significant differences that show variations between goods or services. Otherwise, if the scales are statistically reliable and valid but do not differentiate between different destinations, it can be inferred that there may be opinions that bring bias to the results.

The study carried out has shown that two destinations, Gran Canaria and Tenerife, are evaluated by customers who are similar. In addition, customers also see significant differences between these two destinations and the accommodations offered in Agadir. In this context, it can be seen that, although Agadir is located relatively close to the Canary Islands and also focuses on sun and beach tourism, it presents significant differences on practically all the variables analyzed when compared to Gran Canaria and Tenerife.

In conclusion, this paper has developed a methodology to determine the degree of convergence or divergence in online social communication about tourist lodgings and destinations. This methodology has been shown to be a valid way to determine whether the online reputation created by the social communication process generates information that allows potential customers to choose among the various alternatives available to make a purchase decision. Along these lines, the results demonstrated that the data provided by the Booking.com website fulfills this characteristic.

The practical and research implications of this study are various. First, the competitiveness of the accommodation is assessed. Second, this methodology can be applied for specific establishments in relation to their competitors in order to determine their positioning in the market. Third, it is a means of carrying out studies of competing destinations since price, added value, and perceived service quality are essential factors for customer decision-making. Fourth, surveys of areas of the same destination can be conducted to determine the strengths and weaknesses of the overall online reputation. Lastly, depending on the different results obtained, corrective actions can be carried out both internally and via the Internet in order to improve the image of the accommodation and destinations.

It should be noted that this study was carried out only with data from Booking.com, which means that the information being validated in this study is exclusively related to online social communication transmitted through this portal. Future research should evaluate whether other specialized websites have the same capacity to differentiate or assimilate and, consequently, validate the information available on the websites. It is a major challenge to develop methods that can serve to verify the degree of accuracy and consistency of the online social communication constantly taking place between users. It is also necessary to detect whether intentional opinions are expressed in order to guide the online reputation of accommodations and destinations toward a higher or lower value than the real one. Therefore, one of the main challenges of research in online social communication is to establish methodologies to determine its degree of reliability and validity.

**Author Contributions:** The authors have contributed equally in the research design and development, the data analysis, and the writing of the paper. The authors have read and approved the final manuscript.

**Funding:** This research received no external funding.

**Conflicts of Interest:** The authors declare no conflicts of interest.

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
