# Peer review of "Analysis of the Online Reputation Based on Customer Ratings of Lodgings in Tourism Destinations"

_admsci, doi:10.3390/admsci8030051_

Round 1

Reviewer 1 Report

I have read this manuscript twice and enjoyed. I have been involved into the relevant research (other areas, other purposes) myself, and I can say that this article is what I have dreamed about before to have for general reference. Undoubtedly, this is a great contribution to the tourism theory and methodology. I can also add that this paper is well-written.

I have only two principal suggestions:

1) I strongly encourage the author(s) to include the section Discussion. Validity of results, comparisons, etc. can be presented there. This will also make the paper more appealing.

2) Some information from what is now Conclusions can be moved to Introduction and Discussion. Conclusions should be short, whereas general remarks and additional explanations should take correct place in this paper.

Following these recommendations will require minor-to-moderate changes, which will not affect the main message of the paper.

I hope to see this paper published soon.

Author Response

Thank you very much for your comments and suggestions that have been very helpful in improving the study. The following is a list of the changes made:

·       In line with your and Reviewer 2 suggestion, the title of the Conclusions section has been changed to Discussion and Conclusions.

·       In the initial part where a general review of the study is made, the aspects that have been demonstrated in the results obtained are highlighted in order to give it a nuance of discussion and conclusion.

·        The discussion of the results has been expanded with reference to comparative analysis. It also includes another paragraph setting out the main practical and research implications.

Reviewer 2 Report

I recommend you to reinforce the literature on web reputation with the following papers:

a.       Inversini, A., Cantoni, L., & Buhalis, D. (2009). Destinations' information competition and web reputation. Information technology & tourism, 11(3), 221-234.

b.      Jeong, M., Oh, H., & Gregoire, M. (2003). Conceptualizing web site quality and its consequences in the lodging industry. International Journal of Hospitality Management, 22(2), 161-175.

c.       Ladhari, R., & Michaud, M. (2015). eWOM effects on hotel booking intentions, attitudes, trust, and website perceptions. International Journal of Hospitality Management, 46, 36-45.

d.      Law, R., & Hsu, C. H. (2005). Customers' perceptions on the importance of hotel web site dimensions and attributes. International Journal of Contemporary Hospitality Management, 17(6), 493-503.

e.      Mauri, A. G., & Minazzi, R. (2013). Web reviews influence on expectations and purchasing intentions of hotel potential customers. International Journal of Hospitality Management, 34, 99-107.

f.        Micera, R., & Crispino, R. (2017). Destination web reputation as “smart tool” for image building: The case analysis of naples city-destination. International Journal of Tourism Cities, 3(4), 406-423.

g.       Rong, J., Li, G., & Law, R. (2009). A contrast analysis of online hotel web service purchasers and browsers. International Journal of Hospitality Management, 28(3), 466-478.

h.     Wahab, O. A., Bentahar, J., Otrok, H., & Mourad, A. (2015). A survey on trust and reputation models for Web services: Single, composite, and communities. Decision Support Systems74, 121-134.

I recommend also to insert in the introduction, at the end, the structure of the paper.

In the methodology, it is not clear if the first formula AVGH has a scientific foundation. Lack a reference.

Finally, it is more important to improve the conclusion, adding discussion (rename the paragraph in “Discussion and conclusion”).

In this paragraph it is necessary to consider results and recall scientific literature on the basis of the work. Moreover, it is necessary to highlight theoretical implication and practical implication.

Author Response

Thank you very much for your comments and suggestions that have been very helpful in improving the study. The following is a list of the changes made:

·       A paragraph has been included at the end of the introduction setting out the structure of the document.

·       The literature review has been expanded to include suggested citations and some more that have been included.

·       A reference is included in the AVGH formula.

·       A final section entitled Discussion and Conclusions has been created, including references to the literature, broadening the discussion of the results obtained and exposing the theoretical and practical implications.

Round 2

Reviewer 2 Report

I suggest to see also Micera, Crispino 2017 for the literature review and to add the structure of paper in  the introduction.

Author Response

Thank you very much for the comments that have been taken into account in including a paragraph at the end of the introduction with the structure of the article. On the other hand, the quotation from Micera and Crispino 2017 was expanded in the literature review.